

# The health care and life sciences community profile for dataset descriptions

Michel Dumontier[1,*], Alasdair J.G. Gray[2,*], M. Scott Marshall[3,*], Vladimir Alexiev[4], Peter Ansell[5], Gary Bader[6], Joachim Baran[1], Jerven T. Bolleman[7], Alison Callahan[1], José Cruz-Toledo[8], Pascale Gaudet[9], Erich A. Gombocz[10], Alejandra N. Gonzalez-Beltran[11], Paul Groth[12], Melissa Haendel[13], Maori Ito[14], Simon Jupp[15], Nick Juty[15], Toshiaki Katayama[16], Norio Kobayashi[17], Kalpana Krishnaswami[18], Camille Laibe[15], Nicolas Le Novère[19], Simon Lin[20], James Malone[15], Michael Miller[21], Christopher J. Mungall[22], Laurens Rietveld[23], Sarala M. Wimalaratne[15] and Atsuko Yamaguchi[16]

[1] Stanford Center for Biomedical Informatics Research, Stanford University, Stanford, CA, United States of America
[2] Department of Computer Science, Heriot-Watt University, Edinburgh, United Kingdom
[3] Department of Radiation Oncology (MAASTRO), GROW— School for Oncology and Developmental Biology, MAASTRO Clinic, Maastricht, Netherlands
[4] Ontotext Corporation, Sofia, Bulgaria
[5] CSIRO, Australia
[6] The Donnelly Centre, University of Toronto, Toronto, Canada
[7] Swiss-Prot group, SIB Swiss Institute of Bioinformatics, Geneve, Switzerland
[8] Carleton University, Canada
[9] CALIPHO group, SIB Swiss Institute of Bioinformatics, Geneve, Switzerland
[10] IO Informatics, Berkeley, CA, United States of America
[11] Oxford e-Research Centre, University of Oxford, Oxford, Oxfordshire, United Kingdom
[12] Elsevier Labs, Netherlands
[13] Department of Medical Informatics and Epidemiology, Oregon Health Sciences University, Portland, OR, United States of America
[14] Office of Medical Informatics and Epidemiology, Pharmaceuticals and Medical Devices Agency, Chiyoda-ku, Japan
[15] EMBL, European Bioinformatics Institute, Saffron Walden, United Kingdom
[16] Database Center for Life Science, Kashiwa, Japan
[17] Advanced Center for Computing and Communication, RIKEN, Wako-shi, Saitama, Japan
[18] Cerenode Inc., United States of America
[19] The Babraham Institute, Cambridge, United Kingdom
[20] Nationwide Children's Hospital, Columbus, OH, United States of America
[21] Institute for Systems Biology, Seattle, WA, United States of America
[22] Environmental Genomics and Systems Biology, Lawrence Berkeley National Laboratory, Berkeley, CA, United States of America
[23] Department of Exact Sciences, VU University Amsterdam, Amsterdam, Netherlands
* These authors contributed equally to this work.

Corresponding authors
Michel Dumontier,
micheldumontier@gmail.com,
michel.dumontier@stanford.edu
Alasdair J.G. Gray,
A.J.G.Gray@hw.ac.uk
M. Scott Marshall,
mscottmarshall@gmail.com

## ABSTRACT

Access to consistent, high-quality metadata is critical to finding, understanding, and reusing scientific data. However, while there are many relevant vocabularies for the annotation of a dataset, none sufficiently captures all the necessary metadata. This

prevents uniform indexing and querying of dataset repositories. Towards providing a practical guide for producing a high quality description of biomedical datasets, the W3C Semantic Web for Health Care and the Life Sciences Interest Group (HCLSIG) identified Resource Description Framework (RDF) vocabularies that could be used to specify common metadata elements and their value sets. The resulting guideline covers elements of description, identification, attribution, versioning, provenance, and content summarization. This guideline reuses existing vocabularies, and is intended to meet key functional requirements including indexing, discovery, exchange, query, and retrieval of datasets, thereby enabling the publication of FAIR data. The resulting metadata profile is generic and could be used by other domains with an interest in providing machine readable descriptions of versioned datasets.

## INTRODUCTION

Big Data presents an exciting opportunity to pursue large-scale analyses over collections of data in order to uncover valuable insights across a myriad of fields and disciplines. Yet, as more and more data is made available, researchers are finding it increasingly difficult to discover and reuse these data. One problem is that data are insufficiently described to understand what they are or how they were produced. A second issue is that no single vocabulary provides all key metadata fields required to support basic scientific use cases (*Ohno-Machado et al., 2015*). A third issue is that data catalogs and data repositories all use different metadata standards, if they use any standard at all, and this prevents easy search and aggregation of data (*Vasilevsky et al., 2013*). To overcome these challenges, we have come together as a community to provide guidance for defining essential metadata to accurately describe a dataset, and the manner in which we can express it. The resulting descriptions support the publication of FAIR datasets that are Findable, Accessible, Interoperable, and Reusable (*Wilkinson et al., 2016*).

For the purposes of this article, we reuse the definition of a dataset from (*Maali & Erickson, 2014*). That is, a dataset is defined as

A collection of data, available for access or download in one or more formats. For instance, a dataset may be generated as part of some scientific investigation, whether tabulated from observations, generated by an instrument, obtained via analysis, created through a mash-up, or enhanced or changed in some manner. Research data is available from thousands of speciality databases listed in the Nucleic Acids Research database issue (http://www.oxfordjournals.org/nar/database/paper.html, accessed June 2016), in specialized data archives such as those hosted by the EBI (http://www.ebi.ac.uk/, accessed June 2016) and the NCBI (http://www.ncbi.nlm.nih.gov/, accessed June 2016), in literature curated databases such as ChEMBL (https://www.ebi.ac.uk/chembl/, accessed June 2016), (*Bento et al., 2014*) PharmGKB (https://www.pharmgkb.org/, accessed June 2016) or the CTD (http://ctdbase.org/, accessed June 2016), or from research repositories such as

BioMedCentral-BGI GigaScience (http://www.gigasciencejournal.com/, accessed June 2016), Nature Publishing Group's Scientific Data (http://nature.com/sdata/, accessed June 2016), Dryad Digital Repository (http://datadryad.org/, accessed June 2016), FigShare (http://figshare.com/, accessed June 2016), and Harvard Dataverse (https://dataverse.harvard.edu/, accessed June 2016). Cross-repository access is possible through data catalogs such as Neuroscience Information Framework (NIF) (http://www.neuinfo.org/, accessed June 2016), BioSharing (http://biosharing.org/, accessed June 2016), Identifiers.org Registry (http://identifiers.org/, accessed June 2016), Integbio Database Catalog (http://integbio.jp/dbcatalog/, accessed June 2016), Force11 (https://www.force11.org/catalog, accessed June 2016), and CKAN's datahub (http://datahub.io/, accessed June 2016).

While several vocabularies are relevant in describing datasets, none are sufficient to completely provide the breadth of requirements identified in Health Care and the Life Sciences. The Dublin Core Metadata Initiative (DCMI) (*DCMI Usage Board, 2012*) Metadata Terms offers a broad set of types and relations for capturing document metadata. The Data Catalog Vocabulary (DCAT) (*Maali & Erickson, 2014*) is used to describe datasets in catalogs, but does not deal with the issue of dataset evolution and versioning. The Provenance Ontology (PROV) (*Lebo, Sahoo & McGuinness, 2013*) can be used to capture information about entities, activities, and people involved in producing or modifying data. The Vocabulary of Interlinked Datasets (VoID) (*Alexander et al., 2011*) is an RDF Schema (RDFS) (*Brickley & Guha, 2014*) vocabulary for expressing metadata about Resource Description Framework (RDF) (*Cyganiak, Wood & Lanthaler, 2014*) datasets. Schema.org (http://schema.org/Dataset, accessed June 2016) has a limited proposal for dataset descriptions. Thus, there is a need to combine these vocabularies in a comprehensive manner that meets the needs of data registries, data producers, and data consumers, i.e., to support the publication of FAIR data.

Here we describe the results of a multi-stakeholder effort under the auspices of the W3C Semantic Web for Health Care and Life Sciences Interest Group (HCLS) (http://www.w3.org/blog/hcls/, accessed June 2016) to produce a specification for the description of datasets that meets key functional requirements, uses existing vocabularies, and is expressed using the Resource Description Framework. We discuss elements of data description including provenance and versioning, and describe how these can be used for data discovery, exchange, and query (with SPARQL (*The W3C SPARQL Working Group, 2013*)). This then enables the retrieval and reuse of FAIR data to encourage reproducible science.

The contributions of this paper are:

- A description of the process followed to generate the community profile for dataset descriptions (given in the Methods Section);
- A summary of the community profile (given in the Results Section), a full specification is provided in the W3C Interest Group Note (*Gray et al., 2015*);
- An analysis of where the community profile fits with existing efforts for providing dataset descriptions (given in the Discussion Section).

## METHODS

### The health care and life sciences community group

The health care and life sciences community profile for describing datasets was developed as a collaborative effort within the Health Care and Life Sciences Interest Group (HCLSIG) of the World Wide Web Consortium (W3C) (http://w3.org, accessed June 2016). This group has a mission to support and develop the use of Semantic Web technology across the health care and life sciences domains. Membership of the interest group is drawn from research and industry organisations from all over the world. A self-selecting subset of these members, i.e., those with an interest in the community project, were actively involved in the discussions for the community profile. Interested individuals could take part in the process via the weekly telephone conferences that were advertised on the HCLSIG mailing list, email discussions on the HCLSIG mailing list, or commenting directly on the working drafts of the various documents or raising issues on the issue tracker. The initial stages of the work saw high levels of engagement with later stages only seeing the core team involved. However, in general it was during the early stages that most of the community agreement was made and the later stages were devoted to writing up the process.

### Developing the profile

The purpose of developing a community profile was to promote the discovery of datasets, enable their reuse, and tracking the provenance of this reuse. An overarching goal in the development of the community profile was to identify and reuse existing vocabulary terms rather than create yet another vocabulary for describing datasets. We believe that this approach will enable greater uptake of the profile due to the existing familiarity with many of these terms.

The development of the community profile was driven by use cases that were collected from the interest group members. Each member of the group was encouraged to supply details of the use cases they would like to support with a community profile for dataset descriptions. A total of 15 use cases were supplied covering data consumption, data publishing, and data cataloguing. The full text of these use cases can be found in Section 9 of the W3C Interest Group Note (https://www.w3.org/TR/hcls-dataset/#usecases, accessed June 2016) (*Gray et al., 2015*), the text of which has been supplied as supplemental material.

The use cases were analysed for common usage patterns for metadata in order to identify the metadata properties that would be required. In addition to the common metadata properties found in most dataset description vocabularies, the use cases identified requirements to support:

1. Distinct resolvable identifiers for the metadata about a dataset, its versions, and the distributions of these versions;
2. Descriptions of the identifiers used within a resource;
3. Details of the provenance of the data;
4. Rich statistics about RDF data to support the querying of a SPARQL endpoint.

The identified requirements fed into the second activity of the group.

The second strand of activity was to collect existing vocabulary terms for each of the desired properties. This included analysing existing dataset metadata publication practices
(see the Discussion Section for details) as well as identifying other potential terms using tools such as the Linked Open Vocabulary repository (http://lov.okfn.org/, accessed June 2016) (*Vandenbussche & Vatant, 2014*), BioPortal (http://bioportal.bioontology.org/, accessed June 2016) (*Whetzel et al., 2011*), and Web search engines.

Over the course of several months, the community group discussed each property for its inclusion in the community profile. An important consideration in this process was the set of possible semantic consequences for each choice, e.g., usage of the Vocabulary of Interlinked Dataset properties would entail that the dataset was available as RDF which is not true of all health care and life sciences datasets. A vote was conducted for each property to choose the appropriate term for the property and its level of requirement. The requirement level was specified with terms such as MUST, SHOULD, or MAY, as defined by RFC 2119 (*Bradner, 1997*).

## Process management

During the development of the community profile, a variety of document management and discussion approaches were used. The reason for using different approaches was due to the affordances that they offered.

During the requirement capture phase it was important to support as many people contributing their use cases and vocabulary terms as possible. We decided that there should be as low an entry barrier to participation as possible. This phase of the development was conducted through a shared and open Google document (https://docs.google.com/document/d/1zGQJ9bO_dSc8taINTNHdnjYEzUyYkbjglrcuUPuoITw/edit?usp=sharing, accessed June 2016) and spreadsheet (https://docs.google.com/spreadsheets/d/1bhbw1HAp5I_c9JvAxyURKW0uEGJ8jV5hlzf07ggWDxc/edit?usp=sharing, accessed June 2016) respectively. These enabled multiple participants to concurrently edit the document using a Web browser in an interface similar to the corresponding desktop application. It is possible to comment on sections of text and discuss issues through threading of comments in the context of the document, as well as to have live chat sessions in the margins of the document Web page while editing. The majority of the community profile was developed within the shared Google document.

In preparation for publishing the community profile as a W3C Interest Group Note, the content of the Google document was transformed into a HTML document and stored in an open GitHub repository (https://github.com/W3C-HCLSIG/HCLSDatasetDescriptions, accessed June 2016). At this point the Google document was made read-only, to avoid missing edits, and a link to the new location inserted at the top. The GitHub repository allowed for the continued collaborative editing of the document, although not through a WYSIWIG online editor or in the same interactive collaborative way. However, the GitHub repository enabled better management of the HTML version as well as tracking and responding to issues relating to the document. A real-time preview of the editors' draft version of the interest group note was available using the GitHub HTML preview feature (See http://htmlpreview.github.io/?https://github.com/W3C-HCLSIG/HCLSDataset Descriptions/blob/master/Overview.html for an example (accessed June 2016)). It was this preview location that was circulated on the mailing list and used in discussions during the telephone conferences.

In accordance with W3C procedures, once the community profile was finalised, the preview version was circulated via the W3C HCLSIG mailing list with a link to the issue tracker for generating new issues. Once these final issues were resolved, the HCLSIG formally voted to accept the community profile during a telephone conference in April 2015. The note was then published on the W3C pages in May 2015 once styling issues had been resolved.

## RESULTS

We developed a community profile for the description of a dataset that meets key functional requirements (dataset description, linking, exchange, change, content summary), reuses 18 existing vocabularies, and is expressed in a machine readable format using RDF (*Cyganiak, Wood & Lanthaler, 2014*). The specification covers 61 metadata elements pertaining to data description, identification, licensing, attribution, conformance, versioning, provenance, and content summary. For each metadata element a description and an example of its use is given. Full details of the specification can be found in the W3C Interest Group Note (*Gray et al., 2015*). Here, we will summarise the features of the community profile.

The community profile extends the DCAT model (*Maali & Erickson, 2014*) with versioning through a three component model (Fig. 1), and detailed summary statistics. The three components of the dataset description model are:

**Summary Level Description:** provides a description of the dataset that is independent of file formats or versions of the dataset. For example, this level will capture the title of the dataset which is not expected to change from one version of the dataset to another, but will not contain details of the version number. This is akin to the information that would be captured in a dataset registry.

**Version Level Description:** provides a description of the dataset that is independent of the file formats but tied to the specific release version of a dataset. For example, this level will capture the release date and version number of a specific version of the dataset but will not contain details of where the data files can be obtained.

**Distribution Level Description:** provides a description of the files through which a specific version of a dataset is made available. Examples of the types of metadata captured are the file format, the location from which it is made available, and summary statistics about the data model (e.g., number of triples in the RDF distribution).

Each description component has a different set of metadata properties specified at the appropriate requirement level—mandatory (MUST), recommended (SHOULD), and optional (MAY).

### Modular approach

The community profile is split into five thematic modules, each focusing on a different aspect of the metadata. However, this is simply to ease the presentation and understanding of the properties within the specification. The properties covered in each module must be supplied to provide a conformant description of a dataset. The modules and their focus are as follows.

**Core Metadata:** captures generic metadata about the dataset, e.g., its title, description, and publisher.

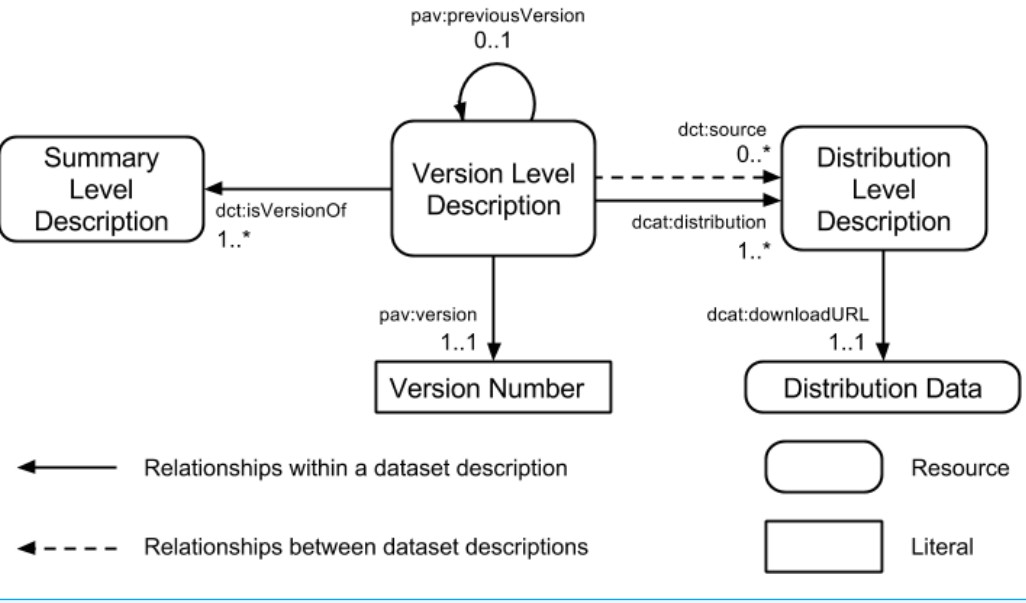

**Figure 1** Three component model for dataset description.

**Identifiers:** describes the patterns used for identifiers within the dataset and for the URI namespaces for RDF datasets.

**Provenance and Change:** describes the version of the dataset and its relationship with other versions of the same dataset and related datasets, e.g., an external dataset that is used as a source of information.

**Availability/Distributions:** provides details of the distribution files, including their formats, in which the dataset is made available for reuse.

**Statistics:** used to summarise the content of the dataset.

Note that in the current specification the statistics presented only make sense for RDF distributions of the data. However, it is in this case that these play the most important role since they provide a summary that will enable others to more effectively assess the contents of the data, as well as query the data. Indeed, the statistical summary matches the results of the queries used to explore a new SPARQL endpoint, e.g., providing the total number of triples, the number of distinct subjects and objects, and the number of relationships between each subject and object pair. Another motivation for providing these statistics in the dataset description is due to the fact that many of the required queries are computationally demanding on the data provider. Thus it is better to execute them once and publish the results in the distribution level description. For other distributions, such as a relational database dump, the data generally provides a schema to describe the relationships in the data.

## Vocabulary reuse

As stated in the Methods Section, a goal of the community profile was to reuse existing vocabulary terms. This was broadly possible except in three cases, which we will now discuss along with our chosen solutions.

To capture the link between a Summary Level Description and the current version of a dataset, a new vocabulary term was required. No existing term could be found to provide this information. Other versioning related properties such as the version number or the relationship to the previous version were supplied using the Provenance, Authoring and Versioning Vocabulary (PAV) (*Ciccarese et al., 2013*). We engaged the creators of the PAV ontology to have them create the property `pav:hasCurrentVersion` as we felt that this was an omission from the original vocabulary and would be beneficial to the wider community. After discussing the use cases where this was required, the PAV authors added the term to their vocabulary.

The second case involved the description of identifiers and their relationship to namespaces used in the dataset. Identifiers.org (*Juty, Le Novère & Laibe, 2012*) is a key resource that provides metadata about life science databases and their identifier schemes. We worked with the Identifiers.org team to develop the idot vocabulary to include terms to define primary short names (`idot:preferredPrefix`), alternate short names (`idot:alternatePrefix`), core identifier patterns (`idot:identifierPattern`) and example identifiers (`idot:exampleIdentifier`). We leverage the VoID vocabulary to specify full URI templates (`void:uriRegexPattern`) and URI identifiers (`void:exampleResource`).

The final case was the ability to capture the relationship between a resource in an RDF dataset and the description of the distributions of the dataset that contain it. On the surface, the `void:inDataset` property would seem to convey this information. However, the domain of this property is `foaf:Document`, not the resource. This is due to the linked data assumptions inherent in VoID where triples are collected and served as documents on the Web. Since VoID is no longer actively developed or maintained, it was decided that a new term should be created in the Semanticscience Interlinked Ontology (*Dumontier et al., 2014*). The new terms, `sio:has-data-item` and `sio:is-data-item-in`, can be used to capture the relationship between a resource and the RDF distributions that it is available in.

## Implementations

A worked example using the ChEMBL dataset (*Bento et al., 2014*) as an exemplar is provided in the Interest Group Note (*Gray et al., 2015*), and included in the supplementary material for this article (hcls-chembl-example.ttl.txt); a summary of the ChEMBL example is given in Table 1. Figure 2 shows an excerpt of this file providing the summary level description for the ChEMBL dataset. The provided description is not intended as an accurate description of the ChEMBL dataset, but an illustration of how to provide the different description levels of the community profile and the various properties that should be used at each of the description levels. In particular, the example provides a sample usage, at each description level, of each of the 61 properties that may be included. However, these are not complete, e.g., the ChEMBL dataset has many authors but only one has been declared in the example.

Table 1 shows that only a few properties are required at each description level. We note that many of these are repeated from one level to another to enable each level to be as self-contained as possible. For example, various core metadata properties such as publisher, license, and rights are kept the same on all levels of the description. The RDF distribution

**Table 1 Summary of the resources in the ChEMBL example dataset description.**

| Resource | Description | Number of triples |
|---|---|---|
| `:chembl` | Summary level description of the ChEMBL dataset | 23 |
| `:chembl17` | Version level description corresponding to version 17 of the ChEMBL dataset | 42 |
| `:chembl17db` | Distribution level description corresponding to an SQL dump of the ChEMBL17 database | 48 |
| `:chembl17rdf` | Distribution level description corresponding to an RDF release of the ChEMBL17 database in the turtle serialisation | 107 |

description is significantly larger due to the inclusion of an example of each of the types of statistical information provided. However, these are automatically generated using the SPARQL queries given in the Interest Group Note. Thus, they can be generated as part of the data publishing pipeline used to create the dataset and its metadata description.

We note that separate distribution level descriptions need to be provided for each RDF serialisation. However, since we expect that the dataset descriptions are generated automatically as part of the data publishing pipeline, there is no additional human effort required. The advantage is that the provenance of dataset reuse can be captured at the level of the distribution format used and thus problems identified in the data can be more effectively tracked. For example, the discrepancy could be a result of a problem in the transformation script that generates the N-triples serialisation of a dataset.

The HCLSIG is currently evaluating the specification with implementations for dataset registries such as Identifiers.org (*Juty, Le Novère & Laibe, 2012*) and Riken MetaDatabase (http://metadb.riken.jp/metadb/front, accessed June 2016) and Linked Data repositories such as Bio2RDF (*Callahan et al., 2013*) and the EBI RDF Platform (*Jupp et al., 2014*). Another example implementation is provided for PHI-Base[1] (*Rodríguez Iglesias et al., 2016*).

## Validating dataset descriptions

To help encourage uptake of the community profile, we have developed an online validation tool to identify when a dataset description conforms to the community profile (*Baungard Hansen et al., 2016*). The online version of the tool enables the user to paste their dataset description in a variety of RDF serialisations. It then analyses the RDF, identifies the resources that provide dataset descriptions and validates the properties expressed against those expected for that level of description. The user can override the suggested description level, e.g., if the tool suggests that a resource is a summary level description but the user wants to validate it as a version level description they can specify that. The validator will either report a successful validation with a green tick or provide contextualised details of where errors have occurred. This allows dataset publishers to verify their descriptions.

The validation tool is also available for download (https://github.com/HW-SWeL/ShEx-validator, accessed June 2016) and using the node.js framework (https://nodejs.org/, accessed June 2016) can be incorporated into data publishing pipelines. Thus, validation of the dataset description can become an integral part of data publishing.

[1] Location of the Summary Level Description for the semantic version of PHI-Base http://linkeddata.systems/SemanticPHIBase/Metadata accessed June 2016

```
BASE <http://rdf.ebi.ac.uk/chembl/>
PREFIX : <http://rdf.ebi.ac.uk/chembl/>
PREFIX ncit: <http://ncicb.nci.nih.gov/xml/owl/EVS/Thesaurus.owl#>
PREFIX skos: <http://www.w3.org/2004/02/skos/core#>

PREFIX cito: <http://purl.org/spar/cito/>
PREFIX dcat: <http://www.w3.org/ns/dcat#>
PREFIX dctypes: <http://purl.org/dc/dcmitype/>
PREFIX dct: <http://purl.org/dc/terms/>
PREFIX foaf: <http://xmlns.com/foaf/0.1/>
PREFIX freq: <http://purl.org/cld/freq/>
PREFIX idot: <http://identifiers.org/idot/>
PREFIX lexvo: <http://lexvo.org/ontology#>
PREFIX pav: <http://purl.org/pav/>
PREFIX prov: <http://www.w3.org/ns/prov#>
PREFIX rdf: <http://www.w3.org/1999/02/22-rdf-syntax-ns#>
PREFIX rdfs: <http://www.w3.org/2000/01/rdf-schema#>
PREFIX schemaorg: <http://schema.org/>
PREFIX sd: <http://www.w3.org/ns/sparql-service-description#>
PREFIX sio: <http://semanticscience.org/resource/>
PREFIX xsd: <http://www.w3.org/2001/XMLSchema#>
PREFIX void: <http://rdfs.org/ns/void/>
PREFIX void-ext: <http://ldf.fi/void-ext#>

###Summary Level (Complete)
:chembl
    rdf:type dctypes:Dataset;
    dct:title "ChEMBL"@en ;
    dct:alternative "ChEMBLdb"@en ;
    dct:description """ChEMBL is a database of bioactive compounds, their quantitative
        properties and bioactivities (binding constants, pharmacology and ADMET, etc).
        The data is abstracted and curated from the primary scientific literature."""@en ;
    dct:publisher :ebi ;
    foaf:page <http://www.ebi.ac.uk/chembl/> ;
    schemaorg:logo <http://www.ebi.ac.uk/rdf/sites/ebi.ac.uk.rdf/files/resize/images/rdf/chembl_service_logo-146x48.gif> ;
    dct:license <http://creativecommons.org/licenses/by-sa/3.0/> ;
    dct:rights """The data in ChEMBL is covered by the Creative Commons By Attribution.
        Under the -BY clause, we request attribution for subsequent use of ChEMBL. For
        publications using ChEMBL data, the primary current citation is:

        A. Gaulton, L. Bellis, J. Chambers, M. Davies, A. Hersey, Y. Light, S. McGlinchey,
        R. Akhtar, A.P. Bento, B. Al-Lazikani, D. Michalovich, & J.P. Overington (2012)
        'ChEMBL: A Large-scale Bioactivity Database For Chemical Biology and Drug Discovery'
        Nucl. Acids Res. Database Issue. 40 D1100-1107 DOI:10.1093/nar/gkr777 PMID:21948594

        If ChEMBL is incorporated into other works, we ask that the ChEMBL IDs are
        preserved, and that the release number of ChEMBL is clearly displayed."""@en ;
    dcat:theme ncit:C48807 ; #chemical
    dcat:keyword "assay"^^xsd:string, "chemical"^^xsd:string ;
    dct:references <http://dx.doi.org/10.1093/bioinformatics/btt765> ;
    rdfs:seeAlso <http://en.wikipedia.org/wiki/ChEMBL> ;
    cito:citesAsAuthority <http://nar.oxfordjournals.org/content/40/D1/D1100> ;
    dct:hasPart :chembl17_rdf_molecule_dataset, :chembl17_rdf_target_dataset ;
#Identifiers
    idot:preferredPrefix "chembl" ;
    idot:alternatePrefix "chembldb" ;
#Provenance and Change
    pav:hasCurrentVersion :chembl17 ;
    dct:accrualPeriodicity freq:quarterly;
#Availability/Distributions
    dcat:accessURL <ftp://ftp.ebi.ac.uk/pub/databases/chembl/ChEMBLdb> ;
    void:sparqlEndpoint <https://www.ebi.ac.uk/rdf/services/chembl/sparql>;
.
```

**Figure 2  Example Summary Level description for the ChEMBL database.** The full example is available in the Supplemental Information.

## DISCUSSION

### Existing vocabularies

A number of existing vocabularies have been developed for describing datasets. However it was only by using a combination of multiple vocabularies that we were able to satisfy the use cases identified within the HCLSIG. We will now discuss why some prominent vocabularies in the area were insufficient to meet the needs of the HCLS community.

The Dublin Core Metadata Initiative (DCMI) publish the Dublin Core Terms and Dublin Core Types ontologies (*DCMI Usage Board, 2012*). These are widely used for providing metadata about web resources. They provide a core set of metadata properties such as `dct:title` for the title, `dct:license` for declaring the license, and `dct:publisher` for the publisher. However, there is no prescribed usage of Dublin Core terms, i.e., each and every resource is free to pick and choose which properties to use, nor does it cover all of the properties deemed necessary by the HCLSIG. In order to support the reuse of datasets, it was deemed important that there was a prescribed set of properties that would appear. Additionally Dublin Core does not provide properties to describe identifiers within a resource (since the focus of Dublin Core is on documents not datasets), only supports rudimentary provenance tracking, and does not provide statistics about a dataset.

The community profile recommends using 18 Dublin Core Terms in conjunction with properties drawn from other vocabularies to augment its coverage. This reuse is due to their suitability and existing wide-spread usage.

The Dataset Catalog Vocabulary (DCAT) (*Maali & Erickson, 2014*) is a W3C Recommendation that was developed in the eGovernment Interest Group (http://www.w3.org/egov/, accessed June 2016) and turned into a recommendation by the Government Linked Data Working Group (http://www.w3.org/2011/gld/wiki/Main_Page, accessed June 2016). The goal of the vocabulary is to support the exchange of metadata records between data catalogs. The DCAT specification prescribes the properties that should appear, with many of these drawn from the Dublin Core ontologies. DCAT also created new terms which are used to distinguish between the catalog record of a dataset and the distribution files of the dataset. However, it does not distinguish between different versions of a dataset. Thus, the HCLS community profile extends this two tier model into a three tier model where the versions of a dataset can be distinguished. The extra terms for describing the versioning information in the HCLS community profile are drawn from the Provenance, Authoring and Versioning Vocabulary (PAV) (*Ciccarese et al., 2013*). In addition, the DCAT vocabulary does not support requirements 2–4 but can be supplemented with additional properties.

A popular vocabulary in the Semantic Web community for describing datasets is the Vocabulary of Interlinked Datasets (VoID) (*Alexander et al., 2011*). This vocabulary also adopts many terms from the Dublin Core ontologies, although as with Dublin Core there are no prescribed properties to provide, thus making the use of VoID dataset descriptions more challenging. It was not possible to use VoID as the basis for the HCLSIG community profile as it assumes that all datasets are available as RDF. While the community profile uses RDF to enable dataset descriptions to be machine processable, it does not assume that the dataset that is being described is published in RDF; there are many datasets in the HCLS

domain that do not publish their data in RDF, hence the need for projects like Bio2RDF (*Callahan et al., 2013*).

## FAIR data principles

The FAIR Data Principles outline essential elements to building an ecosystem of digital resources that are Findable, Accessible, Interoperable, and Reusable (*Wilkinson et al., 2016*). The HCLS dataset description guideline and metadata composed using the guideline are FAIR in the following ways:

**Findable:** Data and their metadata can be uniquely identified using HTTP-based URIs meaning that they are globally unique and can be made persistent (F1); data are described using a rich set of metadata elements (F2); metadata contain an identifier for the dataset (F3); the HCLS dataset description guideline can be found as one of the standards listed in BioSharing.org and support datasets becoming findable by providing common properties that can be registered, indexed and searched (F4).

**Accessible:** Metadata can be potentially retrieved using a standardized communications protocol which is open, free, and universally implemented, as well as supporting authentication and authorization, viz. by their HTTP(S) identifiers (A1, A1.1, and A1.2). As the metadata is separated from the data, it is feasible that the metadata remains accessible even if the data is not (A2).

**Interoperable:** Metadata use RDF—a formal, accessible, shared, and broadly applicable language for knowledge representation (I1); metadata use vocabularies (e.g., Dublin Core, DCAT, SIO) that follow FAIR principles (I2); and include qualified references to other datasets (provenance links) and versions of the same dataset (version links) (I3).

**Reusable:** Metadata may feature a rich set of metadata elements (R1) including licensing (R1.1), provenance (R1.2), and conformance to existing community-based vocabularies (R1.3).

Hence our work offers a clear avenue by which providers (researchers, publishers, repositories) can increase the FAIRness of their digital resources.

## Existing community approaches

BioDBCore is a community driven checklist designed to provide core attributes for describing biological databases (*Gaudet et al., 2011*). The BioSharing catalog (https://www.biosharing.org/, accessed June 2016) is a curated and searchable web portal of interrelated data standards, databases, and data policies in the life, environmental, and biomedical sciences; databases are described using the BioDBCore attributes (https://biosharing.org/biodbcore/, accessed June 2016). The databases catalog is progressively populated via in-house curation, assisted by community contributions via two routes: (1) through a collaboration with the Oxford University Press, where information is obtained from the annual Nucleic Acids Research Database issue and Database journal, and (2) via database developers and maintainers, who register their databases. This centralised and curated approach differs from the idea of the community profile described here. We anticipate the data publishers will publish the HCLS metadata descriptions together with their data and where possible embedded within the data. Registries such as BioDBCore can then

harvest these descriptions and BioSharing are working to support automatic submission and updates based on HCLS dataset descriptions, supporting the idea that you write the metadata once and reuse it many times. We note that many of the properties covered in the BioDBCore are included in the HCLSIG community profile, although contact information such as email addresses are not included in the community profile. We anticipate that the usage of ORCID identifiers for individuals (*Haak et al., 2012*) and the ability to dereference these as RDF will eliminate the maintenance problem of keeping contact details up-to-date in many different registries.

The Big Data to Knowledge (BD2K) NIH-funded biomedical and healthCAre Data Discovery Index Ecosystem (bioCADDIE, https://biocaddie.org, accessed June 2016) consortium (*Ohno-Machado et al., 2015*) works to develop the BD2K Data Discovery Index (DDI), to support data discovery complementing PubMed for the biomedical literature. The bioCADDIE Metadata Working Group has produced a metadata specification (*WG3 Members, 2015*) by converging requirements from a set of competency questions collected from the community and a set of existing models, formats and vocabularies used for describing data (including generic ones such as HCLSIG community profile and DataCite, and domain-specific ones such as ISA and SRA-xml). Thus, datasets described using the HCLSIG community profile can be easily included to the bioCADDIE index and work is ongoing at developing a HCLS profile data ingester.

The Open PHACTS Discovery Platform (*Gray et al., 2014*) provides an integrated view over several datasets with rich provenance information provided to identify where each data property originated. To enable this rich provenance, the Open PHACTS project developed their own standard for describing datasets based on a checklist of properties to provide (*Gray, 2013*). The Open PHACTS use case and their standard for dataset descriptions served as useful inputs to the HCLSIG community profile. The HCLSIG profile extends the Open PHACTS approach to support a larger number of use cases.

The Bio2RDF project provides scripts for converting biological datasets from their native format to an RDF representation together with SPARQL endpoints for interrogating and integrating the resulting data (*Callahan et al., 2013*). The conversion process from the original data source to the resulting RDF representation is captured by a provenance record that supplies core metadata about the source and resulting data (https://github.com/bio2rdf/bio2rdf-scripts/wiki/Bio2RDF-Dataset-Provenance, accessed June 2016) and is subsumed by the HCLSIG community profile. A key contribution to the HCLSIG community profile from the Bio2RDF project has been the need for providing rich statistics about the RDF data (https://github.com/bio2rdf/bio2rdf-scripts/wiki/Bio2RDF-Dataset-Summary-Statistics, accessed June 2016) the purpose of which are to support understanding of the dataset without needing to pose common queries—some of which can be expensive to compute. These recommendations have been included in the distribution level descriptions of the HCLSIG community profile.

## CONCLUSIONS

The Health Care and Life Sciences Community Profile for describing datasets has been developed as a community effort to support application needs. The development process has

been inclusive allowing a wide variety of individuals to provide use cases, and to extensively discuss the choices of vocabulary terms made. This effort was enabled through the use of collaborative writing tools. The resulting community profile enables the description of datasets and the different resources they make available: the dataset, its versions, and the distribution files of these versions. The community profile has extended existing best practice for describing datasets in several key areas; most notably enabling the versions and distributions of a dataset to be distinguished and by providing rich statistics about the dataset. These descriptions can be used to publish datasets compliant with the FAIR data principles. While the profile has been developed by the W3C HCLSIG, there are no aspects that are specific to this domain and the guideline is broadly applicable to many different domains and types of digital resources. The community profile is currently undergoing validation by being implemented by several different projects. We anticipate that there will need to be a consolidation period where new use cases are added and improvements made to the existing community profile.

## ACKNOWLEDGEMENTS

We would like to acknowledge the contributions made by all those involved in the development of the W3C Health Care and Life Sciences community profile. In particular we would like to acknowledge Eric Prud'hommeaux the W3C liaison for the Health Care and Life Sciences Interest Group for his contributions in finalising the formatting of the W3C Interest Group Note.

We also acknowledge the BioHackathon series (http://www.biohackathon.org/, accessed June 2016) for providing opportunities to discuss initial ideas for dataset descriptions.

### Funding

Funding for Michel Dumontier was provided in part by grant U54 HG008033-01 awarded by NIAID through funds provided by the trans-NIH Big Data to Knowledge (BD2K) initiative. Alasdair J.G. Gray was partly funded by the Open PHACTS project and Innovative Medicines Initiative Joint Undertaking under grant agreement number 115191, the resources of which are composed of financial contribution from the European Union's Seventh Framework Programme (FP7/2007-2013) and EFPIA companies' in kind contribution. M. Scott Marshall was funded by the European Commission through the EURECA (FP7-ICT-2012-6-270253) project. Gary Bader was supported by the US National Institutes of Health grant (U41 HG006623). Jerven Bollenman's Swiss-Prot group activities are supported by the Swiss Federal Government through the State Secretariat for Education, Research and Innovation. Nicolas Le Novere was funded by the BBSRC Institute Strategic Programme BB/J004456/1. The BioHackathon series is supported by the Integrated Database Project (Ministry of Education, Culture, Sports Science and Technology, Japan), the National Bioscience Database Center (NBDC - Japan), and the Database Center for Life Sciences (DBCLS - Japan). The funders had no role in study design, data collection and analysis, decision to publish, or preparation of the manuscript.

## Grant Disclosures

The following grant information was disclosed by the authors:

NIAID: U54 HG008033-01.

Open PHACTS project and Innovative Medicines Initiative Joint Undertaking: 115191.

US National Institutes of Health grant: U41 HG006623.

Swiss Federal Government.

BBSRC Institute Strategic Programme: BB/J004456/1.

Integrated Database Project.

National Bioscience Database Center (NBDC—Japan).

Database Center for Life Sciences (DBCLS—Japan).

## Competing Interests

Vladimir Alexiev is an employee of Ontotext Corporation, Sofia, Bulgaria; Peter Ansell is an employee of CSIRO, Australia; Erich A Gombocz is an employee of IO Informatics, Berkeley, CA, USA; Paul Groth is an employee of Elsevier Labs, Netherlands; and Kalpana Krishnaswami is an employee of Cerenode Inc. The authors declare there are no competing interests.

## Author Contributions

- Michel Dumontier, Alasdair J.G. Gray and M. Scott Marshall wrote the paper, prepared figures and/or tables, reviewed drafts of the paper, active participation in HCLS Interest Group discussions.
- Vladimir Alexiev, Peter Ansell, Gary Bader, Jerven T. Bolleman, Alison Callahan, José Cruz-Toledo, Pascale Gaudet, Erich A. Gombocz, Alejandra N. Gonzalez Beltran , Paul Groth, Melissa Melissa Haendel, Maori Ito, Simon Jupp, Nick Juty, Toshiaki Katayama, Norio Kobayashi, Kalpana Krishnaswami, Camille Laibe, Nicolas Le Novère, Simon Lin, James Malone, Michael Miller, Christopher J. Mungall, Laurens Rietveld, Sarala M. Wimalaratne and Atsuko Yamaguchi reviewed drafts of the paper, active participation in HCLS Interest Group discussions.
- Joachim Baran prepared figures and/or tables, reviewed drafts of the paper, active participation in HCLS Interest Group discussions.

## Data Availability

The W3C HCLSIG HCLS Dataset Descriptions GitHub repository contains all text and figures associated with the HCLS Dataset Description Guidelines Interest Group Notes: https://github.com/W3C-HCLSIG/HCLSDatasetDescriptions.

## Supplemental Information

Supplemental information for this article can be found online at http://dx.doi.org/10.7717/peerj.2331#supplemental-information.

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
