# Peer review of "The health care and life sciences community profile for dataset descriptions"

_PeerJ, doi:10.7717/peerj.2331_

## Round 0.1 · original submission · Minor Revisions

· Academic Editor

Minor Revisions

Dear Alasdair,

Your manuscript "The health care and life sciences community profile for dataset descriptions " has been assessed by three reviewers. They all thought the findings are valid.

However, they also raised a number of points. A more thorough description of the work, including a expansion of content and method section, and a summary of the use cases with a reference to whatever material was collected as supplemental data will greatly help the paper.

With kind regards,

Qi Liu
Academic Editor, PeerJ

·

Basic reporting

No Comments

Experimental design

No Comments

Validity of the findings

No Comments

Additional comments

As more and more data is produced, it’s a big challenge for researchers to discover and reuse them. The authors of this paper have come together as a community -- the W3C Semantic Web for Health Care and the Life Sciences Interest Group (HCLSIG), to provide guidance for defining essential metadata to accurately describe a dataset. The community produced a specification for the description of datasets that support the publication of FAIR datasets that are Findable, Accessible, Interoperable, and Reusable. The specification covers elements of description, identification, attribution, versioning, provenance, and content summarization. The existing vocabularies were used for describing datasets with extension due to the familiarity with existing terms.
This paper describes the process of developing the profile for dataset descriptions, and gives us a summary of the community profile. The resulting metadata profile is generic and could be used by other domains with an interest in providing machine readable descriptions of versioned datasets. The guidance is necessary in this Big Data era that we can easily find, access, and reuse the scientific data. Although a full specification with examples can be accessible online in the W3C Interest Group Note, As a user, I would like to see a figure showing an example for dataset description.

·

Basic reporting

The submission adheres to PeerJ policies, is written in clear English, and includes a certain amount of background. It is clear how the work fits into the general field of knowledge as a metadata specification for dataset publication. The article structure and figures are appropriate.

However, in my opinion this article does fail to report sufficiently on the actual use cases considered in developing the specification; it does not seem sufficient to rely solely on the acronym "FAIR" as a substitute for reporting the use cases actually collected without any mention of how those use cases might be distilled to one or more of the FAIR elements - although I believe it is probably reasonable to map descriptive metadata to support for the "F" element, "findable"- the authors need to at least say this and reference their use cases. Even a summarization of the use cases with a reference to whatever material was collected as supplemental data (or as *cited data*) would be helpful, and I think is required to have an article that really adequately reports the topic.

Also, prior art vocabularies, to which the authors refer and which they say are insufficient to meet their (not well described) use cases, are not discussed (except briefly, VOID) in terms of how they were represented, amended, extended, invalidated, assembled or superseded in this spec.

The authors do say that they extended DCAT with versioning - but is this all?

A number of very competent informaticians were involved in development of this spec and I would think that a more thorough description of the work product is called for.

Experimental design

This is a report of an engineering proejct to develop a formal metadata spec, so it doesn't include an experiment per se, but the METHOD description is a stand-in for that. I would say that in particular the METHOD section short-changes the reader by omitting any real discussion of use case analysis (see "basic reporting" above).

Validity of the findings

From what I can determine, the dataset description vocabulary produced by the authors is quite sound. However, I am probably much closer to this work than most readers would be, and can take a lot more or less for granted.

Yet for the ordinary reader, the defects in reporting mentioned previously would not allow the reader to convince her- or him-self that the findings were valid, because validity rests on adequacy of the design to the use cases and reasonableness of the engineering tradeoffs - which are not discussed.

Additional comments

I respect the authors involved in this publication, and their work, and I think this is potentially a worthwhile article to write - but wholly incomplete without an expansion of content such as I mention above. As it is, don't think it really says enough.

I do recognize that use case material may be extensive and impossible to fit completely within the page limits or scope of such an article. However, that can be dealt with in several ways. There are various ways use cases could be summarized and discussed in general terms, yet in more detail than was given in this draft. Also, since the use cases were evidently collected, and exist as text somewhere, it would seem to make sense to archive and reference them as supplementary material, to which the authors can refer. Although use cases are not "data" in the common experimental sense, they are for purposes of engineering development. I would like to see that data discussed more, and referenced.

As a less important comment, but still worth mentioning - in lines 52-60 there is nothing mentioned about the standard NCBI and EBI databases, which seems an odd omission. Also I don’t think the FORCE11 catalog page actually belongs with this group as an example.

My recommendation is for a major revision of the manuscript to resolve the issues mentioned above.

·

Basic reporting

The article provides a clear and informative overview of recent work towards metadata standards for dataset discovery in the area of healthcare and the life sciences.

Experimental design

"The submission must describe original primary research within the Scope of the journal.". The work reported may not count as a formal scientific experiment in the sense of investigative primary research; it is rather a supporting activity to enhance the practical impact, discoverability, reproducibility and effectiveness of such efforts. I encourage the editors to publish the work as it deserves a wide audience amongst practitioners.

Validity of the findings

Solid work (although see above w.r.t. 'experiment').

Additional comments

This is important work. I would encourage you to investigate (as follow-up rather than editorial comment on this text) the technical characteristics of W3C's JSON-LD format. The JSON-LD "context" mechanism provides an additional level of indirection that can be used to express at least simple "application profiles" that combine independently maintained RDF vocabularies.

---

## Round 0.2 · accepted · Accept

· Academic Editor

Accept

Since you have answered all the questions the reviewers raised, I am very happy to inform you that your manuscript is accepted.

·

Basic reporting

The authors addressed all my concerns especially now including partial dataset description as a figure in the manuscript with the rest available as supplemental material.

Experimental design

No Comments

Validity of the findings

No Comments

·

Basic reporting

Manuscript adheres to all PeerJ policies, and fully meets basic reporting guidelines.

Experimental design

This article concerns development of an ontology, which from the standpoint of computer science / bioinformatics, is comparable to primary research in other disciplines such as biology. However it is primarily a design activity. I suggest that if PeerJ continues to publish computer science or bioinformatics articles, that the guidelines be modified to explicitly recognize differences between the two activities.

Validity of the findings

Findings are valid and the authors show how they are valid.

Additional comments

The authors have taken on board all editorial comments in a very satisfactory way. I believe this is a high quality and valuable article in its field and recommend acceptance for publication.